# *Astragalus* Polysaccharide Modulates the Gut Microbiota and Metabolites of Patients with Type 2 Diabetes in an In Vitro Fermentation Model

**DOI:** 10.3390/nu16111698

**Published:** 2024-05-30

**Authors:** Xin Zhang, Lina Jia, Qian Ma, Xiaoyuan Zhang, Mian Chen, Fei Liu, Tongcun Zhang, Weiguo Jia, Liying Zhu, Wei Qi, Nan Wang

**Affiliations:** 1College of Biotechnology, Tianjin University of Science and Technology, Tianjin 300457, China; 18737605989@163.com (X.Z.); 15731164636@163.com (L.J.); 18435203547@163.com (Q.M.); tony@tust.edu.cn (T.Z.); qiweismiling@163.com (W.Q.); 2Key Laboratory of Industrial Fermentation Microbiology, Ministry of Education and Tianjin, Tianjin 300457, China; 3Shandong Academy of Pharmaceutical Sciences, Key Laboratory of Biopharmaceuticals, Engineering Laboratory of Polysaccharide Drugs, National-Local Joint Engineering Laboratory of Polysaccharide Drugs, Postdoctoral Scientific Research Workstation, Jinan 2501011, China; gnice@163.com (X.Z.); chenmian@sdaps.cn (M.C.); liufei@sdaps.cn (F.L.); 4The Center of Gerontology and Geriatrics, National Clinical Research Center of Geriatrics, West China Hospital, Sichuan University, Chengdu 610041, China; jiaweiguo@wchscu.cn; 5Institute of Food Science, Zhejiang Academy of Agricultural Sciences, Hangzhou 310021, China; zhuly@zaas.ac.cn

**Keywords:** *Astragalus* polysaccharide, type 2 diabetes mellitus, fecal microbiota, metabolites

## Abstract

This study investigated the effect of astragalus polysaccharide (APS, an ingredient with hypoglycemic function in a traditional Chinese herbal medicine) on gut microbiota and metabolites of type 2 diabetes mellitus (T2DM) patients using a simulated fermentation model in vitro. The main components of APS were isolated, purified, and structure characterized. APS fermentation was found to increase the abundance of *Lactobacillus* and *Bifidobacterium* and decrease the *Escherichia-Shigella* level in the fecal microbiota of T2DM patients. Apart from increasing propionic acid, APS also caused an increase in all-trans-retinoic acid and thiamine (both have antioxidant properties), with their enrichment in the KEGG pathway associated with thiamine metabolism, etc. Notably, APS could also enhance fecal antioxidant properties. Correlation analysis confirmed a significant positive correlation of Lactobacillus with thiamine and DPPH-clearance rate, suggesting the antioxidant activity of APS was related to its ability to enrich some specific bacteria and upregulate their metabolites.

## 1. Introduction

With its incidence increasing year by year, diabetes mellitus has become a common metabolic disease and the third most chronic severe disease after cancer and cardiovascular disease. Diabetes mellitus can be divided into type 1 (T1DM) and type 2 diabetes mellitus (T2DM), with the latter accounting for more than 90% of all diabetic patients. T2DM is not only characterized by hyperglycemia induced by insulin deficiency or resistance [1], but also accompanied by various complications in the heart, blood vessels, kidneys, retina, and nervous system, thus posing a severe risk to human physical and mental health [2]. Acarbose, metformin, and α-glucosidase inhibitors are commonly used chemical hypoglycemic agents with good clinical applications, but they are expensive and have side effects [3], suggesting the need to find a stable, low-side-effect, and low-cost drug or food to improve and treat T2DM.

Intestinal microbiota disorders can lead to various chronic metabolic diseases, including obesity, diabetes, and hyperlipidemia [4]. Dysbiosis of the intestinal microbiota in patients with T2DM has been demonstrated, such as a decrease in the abundance of Firmicutes and short-chain fatty acid-generating bacteria (e.g., *Bifidobacterium*, *Faecalibacterium*, and *Roseburia*), and an increase in the abundance of conditionally pathogenic bacteria (e.g., *Bacteroides caccae*, *Clostridium symbiosum Eggerthella lenta*, and *Escherichia coli*) [5].

Conversely, supplementation with beneficial bacteria can alleviate diabetes. For example, supplementation with *Lactobacillus paracasei* [6] and *Akkermansia muciniphila* [7] improved glucose tolerance and insulin sensitivity in mice. Interestingly, diabetes was also shown to be influenced by the active metabolites of the intestinal microbiota. Specifically, ammonia, some amino acids, and amines (which are usually considered as the end products of protein degradation by harmful bacteria of gut microbiota) are positively associated with T2DM [8]. Additionally, short-chain fatty acids (SCFAs, the main metabolites of intestinal microorganisms) can improve insulin sensitivity and regulate pancreatic insulin secretion in T2DM patients [9]. These reports suggest that regulating intestinal microbiota and its metabolites is a new direction for T2DM treatment.

Plant polysaccharides, with their low toxicity and blood glucose lowering effects, have been a hot research topic in the treatment of T2DM in recent years. For example, *Cyclocarya paliurus* polysaccharide [10], Morus alba polysaccharide [11], and *Dendrobium officinale* polysaccharide [12] have been shown to stabilize glucose and reduce diabetic complications. The dried root of *Astragalus membranaceus* (Fisch.) Bunge (Fabaceae) (Astragali Radix) is one of the key ingredients in many herbal anti-diabetic formulations [13]. *Astragalus* polysaccharide (APS), the main active substance of astragali radix, has been demonstrated to have the function of controlling blood sugar [14], enhancing immunity [15], and antioxidant effects [16]. A recent work has shown that APS could improve glucose regulation in a diabetes mice model by elevating the abundance of *Muribaculum*, *Faecalibaculum*, and *Lactobacillus*, increasing the levels of acetic acid, butyric acid, and propionic acid, and decreasing the expression levels of tumor necrosis factor α (TNF-α) and pro-inflammatory factors interleukin 6 (IL-6) [17]. Similarly, Liu et al. demonstrated that APS stabilized glucose homeostasis in diabetic mice by altering the gut microbiota and regulating the level of short-chain lipoic acid [18]. However, most of these studies are based on animal models such as mice, and the effects of APS on the intestinal microbiota and intestinal metabolic profile in T2DM patients have not been systematically clarified. Therefore, this study aimed to elucidate the regulatory effects of APS on gut microbiota and its metabolic profile in T2DM patients by combining 16S sequencing technology and metabolomics.

## 2. Materials and Methods

### 2.1. Extraction, Purification, and Physicochemical Characterization

Astragali radix was purchased from Min County, Dingxi City, Gansu Province, China. APS was extracted from astragalus radix with boiling water and concentrated by rotary evaporator (Centron Technology Co., Shanghai, China), followed by precipitation in 80% ethanol. Next, the crude polysaccharide was deproteinized by the Sevag method and then filtered through a 3.5 KDa dialysis membrane to remove salt ions, followed by collecting the polysaccharide and lyophilization for further treatment (Marin Christ, Osterode, Germany). The main fractions (APS-0M, APS-0.1M, and APS-0.2M) were obtained from APS by DEAE Sepharose FF cellulose, followed by further isolation and purification of APS-1 and APS-2 from APS-0M and APS-0.2M by Sephadex G-100 column chromatography, respectively (Appendix A). Finally, the carbohydrate, total phenol, and protein contents of APS, APS-1, and APS-2 were assayed by the methods of phenol-sulfuric acid [19], Folin–Ciocalteu [20], and Bradford [21], respectively.

### 2.2. Structural Characterization

For UV-vis assay, APS, APS-1, and APS-2 solutions were scanned in a wavelength range of 360 to 190 nm using an UV spectrophotometer (ND-100C, Miulab, Hangzhou, China). The transmission assay was performed using a Vertex 70 FTIR spectrometer (Bruker, Mannheim, Germany) to collect the spectra of APS, APS-1, and APS-2 between 4000 and 400 cm^−1^.

The molecular weight distribution of APS, APS-1, and APS-2 was analyzed by high-pressure liquid chromatography (HPLC) (Agilent Technologies Inc., Palo Alto, CA, USA) equipped with a refractive index detector and a TSK gel G5000PWXL column (7.8 by 300 mm, 10 μm, TOSOH, Tokyo, Japan). Under the conditions of column temperature 30 °C, maximum column pressure 30 bar, and detector temperature 40 °C, 20 μL of sample was extracted and eluted with ultrapure water as mobile phase at 0.5 mL/min. The standard curve was generated using dextran standards (Mw: 2.7, 5.25, 9.75, 13.05, 36.8, 64.65, 135.35, 300.6, and 2000 kDa) and the relative molecular weight was estimated for each sample.

For monosaccharide analysis, APS, APS-1, and APS-2 were hydrolyzed by the trifluoroacetic acid (TFA) method. Next, the hydrolyzed products of APS, APS-1, and APS-2 were analyzed using Dionex ICS 5000 ion chromatograph (Dionex, Sunnyvale, CA, USA) equipped with a Dionex CarboPac-PA20 analytical column. The retention times and response values of standards were used to draw standard curves and determine the monosaccharide composition and content of samples.

### 2.3. In Vitro Simulated Gastric-Intestinal Digestion

Preparation of simulated gastric buffer (SGF) followed the description in Appendix A, followed by prewarming 4 mL of SGF in a 37 °C water bath, and then supplementation with CaCl_2_ solution (0.3 M, 0.025 mL), pepsin (3000 U/mL, 0.33 mL), polysaccharide solution (8 mg/mL, 5 mL), and ultrapure water to a 10 mL total volume (pH = 3.0). After 2 h of incubation, samples were collected and inactivated, followed by simulated small intestinal digestion. The simulated small intestine buffer (SIF) was configured as shown in Appendix A. After prewarming in a 37 °C water bath, SIF was supplemented with CaCl_2_ solution (0.3 M, 0.1 mL), trypsin (4000 U/mL, 0.25 mL), bile salt solution (100 mg/mL, 0.4 mL), gastric digestive fluid of APS (5 mL), and ultrapure water to a 10 mL total volume (pH = 7.0). After 3 h of incubation, samples were collected and inactivated.

### 2.4. Fermentation In Vitro

Fresh feces for this experiment were provided by fourteen volunteers, including 7 healthy volunteers aged 27 to 58 years (3 females and 4 males) and 7 T2DM volunteers aged 50 to 73 years (3 females and 4 males), who had not taken antibiotics or prebiotics for at least 3 months. All experiments in this study were performed following the Institutional Guidelines of the Biomedical Ethics Committee of West China Hospital of Sichuan University (2018(286)). Each fresh stool was collected and processed into a stool suspension using an automatic stool processor (HALO-P100, Hunan Hailu Biotechnology Co., Changsha, China). Next, the fecal slurry samples were mixed with yeast extract-casein hydrolyzed fatty acid (YCFA) medium or YCFA medium containing 1% (*w*/*v*) APS at 1:10, followed by anerobic incubation at 37 °C for 24 h. In this study, all the samples were divided into 3 groups: in vitro medium-fermented feces of healthy control groups (HC), in vitro medium-fermented feces of T2DM patients (T2DM), and in vitro APS-fermented feces of T2DM patients (APS-T2DM).

### 2.5. Gut Microbiota Analysis

Briefly, the gut microbiota in the fermentation solution were collected by centrifugation at 4 °C and 12,000 rpm for 10 min, followed by bacterial DNA extraction and 16S rDNA sequencing in the Major Biotechnology Co., Ltd. (Shanghai, China). Species sequence comparisons were annotated based on the Ribosome Database Project (RDP). Alpha diversity was analyzed by QIIME software (v1.9.1), and the species diversity of alpha diversity was characterized by Shannon and Simpson indices. Shannon’s index is used to describe the disorder and uncertainty in the occurrence of individuals of a species, the higher the uncertainty, the higher the diversity. Simpson’s index reflects the size of species richness by analyzing the number of individuals in the same population. The value of this index ranges from 0 to 1, with larger values indicating lower species identification and higher concentration of species numbers. It is worth noting that the two indicators are calculated differently and with different emphasis, so they are considered together to avoid differences in the results of individual indicators [16]. Finally, the gut microbiota data were analyzed using the Majorbio Cloud platform (https://cloud.majorbio.com, 13 January 2023).

### 2.6. Metabolomic Analysis

*GC*-*MS.* After centrifugal collection of the supernatant of fermentation samples, 200 µL of supernatant was supplemented with 20 µL of ribitol (1 mg/mL) as an internal standard and then lyophilized in a freeze dryer. Next, the silane derivatization of the samples was performed as previously reported [22]. Gas phase conditions: Agilent HP-5 column (30 m × 0.25 μm × 0.25 mm, Agilent, Santa Clara, CA, USA), helium carrier gas, 1 μL injection volume, and 250 °C injection port temperature. The heating procedure was performed with 50 °C as the initial temperature for 2 min, up to 270 °C at 5 °C/min, 290 °C at 2.5 °C/min, and 310 °C at 10 °C/min and held for 4 min. Mass spectral conditions: 50–700 *m*/*z* scanning range, 230 °C ion source temperature, 280 °C interface temperature, and 10 min solvent delay time.

For HPLC/MS analysis, the samples were pre-treated as follows. After centrifuging the fermentation broth at 10,000 rpm for 15 min, the supernatant was extracted by SPE column, followed by elution with methanol. Next, the eluate was blown dry at 37 °C under nitrogen, followed by dissolution and methanol washing. Meanwhile, the instrument stability during the experiment was assessed with quality control samples (QC, an aliquot mixture of all fermentation samples). Liquid phase conditions: Xbridge C18 column (3.5 μm, 2.1 × 100 mm), 30 °C column temperature, 10 μL injection volume, and 0.25 mL/min flow rate. The mobile phase consisted of 0.1% formic acid water (A) and acetonitrile (B), with the gradient elution program of 10% B at 0–1 min, 10–95% B at 1–12 min, 95% B at 12–15 min, and 10% B at 15–0 min.

Electrospray ionization (ESI) mass spectral analysis was performed using ESI source in a positive ion mode with an Agilent 6230 accurate mass time-of-flight (TOF) mass spectrometer (Santa Clara, CA, USA). The parameters were set as follows: 4000.0 V capillary voltage, 120 °C source temperature, 35 psi nebulizer pressure, 350 °C desolvation temperature, 9 L/min desolvation gas flow rate, 40 V cone voltage, 121.050873 and 922.009798 reference ion *m*/*z*, 175 V collision energy voltage, and helium collision gas.

### 2.7. Quantification of Fecal SCFAs

Briefly, each fermented sample was mixed with crotone-phosphite solution, acidified for 24 h, and filtered into a sample vial. Next, the contents of short-chain fatty acids in each fermented sample (such as acetic, butyric, propionic, isobutyric, isovaleric, and valeric acids) was determined using an Agilent 7890 detector (Santa Clara, CA, USA) with an Agilent FFAP column 30 mm × 0.25 mm × 0.25 μm (Santa Clara, CA, USA). Trans-2-butenoic acid was used as the internal standard for this experiment.

### 2.8. Antioxidant Analysis

Briefly, each sample was mixed with 2,2-diphenyl-1-picrylhydrazyl (DPPH) ethanol solution (0.15 mM) in an equal volume and incubated for 30 min under light-proof conditions. Next, the absorbance value of each solution was measured at 517 nm using a multifunctional enzyme marker (Tecan, Salzburg, Austria). Finally, DPPH removal efficiency is calculated by the following formula:Scavenging effect (%) = [1 − (A _sample 517nm_ − A _control 517nm_)/A _blank517nm_] × 100% 
where A control 517 nm is the absorbance of the control (ethanol instead of DPPH) and A blank 517 nm is the absorbance of the blank (distilled water instead of the samples).

Additionally, 2 mL of each fermentation liquid sample was mixed with 2 mL of FeSO_4_ (6 mM), 2 mL of H_2_O_2_ (6 mM), and 2 mL of sodium salicylate (6 mM), with distilled water used as a positive control. Next, the absorbance value of each reaction product at 510 nm was measured spectrophotometrically. The OH•-scavenging activity was expressed as:%HO• scavenged = (1 − (A _sample 510nm_ − A _control 510nm_)/A _blank510nm_) × 100%

Among them, control 510 nm is the absorbance of control (distilled water replaced sodium salicylate), and blank 510 nm is the absorbance of blank (distilled water replaced the sample).

The reducing power was determined according to Chen et al. [23]. An aliquot of 3 μL of fermentation liquid sample was mixed with 247 μL of phosphate buffer solution (pH 6.6, 0.2 mol/L) and 250 μL 1% potassium ferricyanide (*w*/*v*), and incubated for 30 min at 50 °C. Next, 250 μL 10% trichloroacetic acid (*w*/*v*) was added and centrifuged. Then 100 μL of the supernatant was aspirated and mixed with 100 μL of distilled water and 20 μL 0.1% ferric chloride (*w*/*v*). Finally, these mixtures were analyzed at an absorbance of 700 nm taking distilled water as a reference.

### 2.9. Statistical Analysis

All data were analyzed using SPSS 22 (IBM, New York, NY, USA) and the results were presented as mean ± standard deviation (SD). Data normal distribution and homogeneity of variance were assessed by Shapiro–Wilk and Brown–Forsythe tests, respectively. When the data satisfied normal distribution, differences between three groups were analyzed, and post hoc tests were conducted through one-way ANOVA and Tukey’s test (equal variances) or Dunnet-T3 (unequal variances). If data were not normally distributed, Kruskal–Wallis analysis was performed. Graphs were made with GraphPad Prism 9 software (San Diego, CA, USA). Metabolic data were analyzed using Metaboanalyst 5.0 (https://metaboanalyst.ca, 21 December 2022) and SIMCA 14 (v14.1, Sartorius, Göttingen, Germany). Correlation analysis between differential metabolites and fecal microbiota was performed using Spearman correlation.

## 3. Results

### 3.1. Chemical and Structural Characterization of APS

As shown in Appendix A, the three components of crude APS (APS-0M, APS-0.1M, and APS-0.2M) were separated by DEAE Sepharose FF, with their proportion accounting for 82.72 ± 4.10%, 2.65 ± 0.01%, and 13.14 ± 0.02% of the total polysaccharide, respectively. Finally, APS-1 and APS-2 were further isolated and purified from APS-0M and APS-0.2M, respectively (Appendix A).

Appendix A shows the basic physicochemical properties of APS, APS-1, and APS-2. The yield was about 9.48 ± 0.54% (*w*/*w*), 59.92 ± 0.70% (*w*/*w*), and 32.21 ± 0.20% (*w*/*w*) for APS, APS-1, and APS-2 extracts, and their carbohydrate content was 75.73 ± 0.72% (*w*/*w*), 95.71 ± 1.42% (*w*/*w*), and 92.5 ± 1.25% (*w*/*w*), respectively. Meanwhile, the protein and total phenol content was found to be extremely low in APS, APS-1, and APS-2, with only about 0.08% of proteins found in APS, while no detectable proteins and phenolics in both APS-1 and APS-2, suggesting the virtual absence of proteins and phenolics in APS-1 and APS-2.

In the UV spectra of Figure 1A, APS, APS-1, and APS-2 exhibited no absorption peak at 260 and 280 nm, implying the presence of little or no nucleic acid and protein. In Figure 1B, HPGPC analysis revealed that four significant peaks were observed in APS and the average molecular weight of these four components were about 1928.54 kDa, 396.31 kDa, 239.92 kDa, and 4.87 kDa, respectively. The average molecular weights were about 4.39 and 1915.15 kDa for the purified APS-1 and APS-2, respectively.

In Figure 1C, APS, APS-1, and APS-2 were seen to have similar FT-IR spectra. The strong absorption peak around 3382.97 cm^−1^ can be ascribed to the -OH stretching vibration of sugars, the weak absorption peak around 2923.94 cm^−1^ is attributed to the C–H stretching vibration [24], the weak peaks around 1635.55 and 1411.82 cm^−1^ correspond to the presence of carboxyl groups [25], and the strong absorption peak near 1026.28 is assigned to the presence of a pyran ring in the polysaccharide [26]. The monosaccharide composition in APS was determined by ion chromatography. In Figure 1D, the major monosaccharides were seen to consist of arabinose, galactose, glucose, and galacturonic acid at a molarity ratio of 1:0.33:4.36:0.02 in APS, galactose and glucose at a molar ratio of 1:9.25 in APS-1, and arabinose, galactose, glucose, and galacturonic acid at a molar ratio of 1:0.85:0.96:0.46 in APS-2.

### 3.2. APS Altered the Fecal Microbiota Composition of T2DM Patients

The degradation rate of APS was 1.52% and 2.36% in simulated gastric and simulated intestinal fluids, respectively, indicating that APS can hardly be digested by the stomach and small intestine, so it can be utilized by gut microbes. In addition, Appendix A shows that APS concentration at 50–400 µg/mL had no significant effect on Caco2 activity. Consistent with our results, Ying et al. [27] and Wang et al. [28] also found that APS was almost non-toxic to epithelial cells. In Appendix A, the sequencing depth of samples was seen to sufficiently reflect the composition of most microorganisms in the samples. In Figure 2A,B, the Shannon index was shown to be lower in the T2DM group than in the HC group, in contrast to an opposite result in the Simpson index, despite no significant difference. Meanwhile, APS addition did not significantly affect the Shannon and Simpson indices in the feces of T2DM patients. Further PCoA and PLS-DA analyses confirmed a notable and consistent separation between the groups of HC, T2DM, and APS-T2DM (Figure 2C,D). These results suggest that APS fermentation may have promoted the proliferation of some specific bacteria, but with no effect on microbial diversity.

In Figure 2E, the microbiota of each group at the phylum level were shown to consist mainly of Firmicutes, Bacteroidota, Actinobacteria, and Proteobacteria. Specifically, the T2DM group was lower than the HC group in the relative abundance of Firmicutes, but higher in the relative abundance of Proteobacteria. However, the abundance of Firmicutes and Proteobacteria was significantly reversed by APS fermentation. The different groups of gut microbial taxa were displayed from phylum to genus using LEfSe (Appendix A), with the dominance of Clostridia, Acidaminococcaceae, Lachnospiraceae, and *Phascolarctobacterium* in the HC group, and the dominance of Proteobacteria, Enterobacteriaceae, and *Escherichia-Shigella* in the T2DM group. Meanwhile, Firmicutes and Lactobacillaceae showed significant enrichment in the APS-T2DM group.

The distributions of microorganisms at both family and genus levels are shown in Figure 3A–D. Compared with the HC group, the T2DM group showed higher relative abundances of Enterobacteriaceae, *Escherichia-Shigella*, and *Parabacteroides*. However, the APS addition caused a decrease in the relative abundance of Enterobacteriaceae, *Escherichia-Shigella*, and *Parabacteroides*, and an increase in the relative abundance of *Bifidobacterium* and *Lactobacillus* in T2DM patients, suggesting that APS could regulate the intestinal microbiota structure of T2DM patients.

### 3.3. Metabolic Analysis of APS Fermentation Samples

Appendix A shows the GC-MS and HPLC-MS results of metabolites in fermentation samples, and a total of 113 substances were detected. As shown in Figure 4A, the three groups of samples were clearly separated, indicating the presence of differential metabolites in them. In Figure 4B, the results of orthogonal partial least squared discriminant analysis (OPLS-DA) revealed a significant separation between two pairs in the HC, T2DM, and APS-T2DM groups. All the OPLS-DA models were confirmed to be valid by the permutation test (*n* = 200) (Figure 4C). These results suggest the differences of metabolic profiles among the three groups of samples.

Compounds with significant differences were further screened (Appendix A), and some differential metabolites are shown in Figure 5. The T2DM group was significantly lower than the HC group in butyric acid and biotin and was significantly higher in L-threonine, L-glutamic acid, glycine, and guanine. In contrast with the T2DM group, APS fermentation significantly decreased the levels of L-valine, L-proline, L-threonine and L-glutamic acid, and spermidine, while the levels of all-trans-retinoic acid, glutamine, thiamine, butyric acid, and propanoic acid were elevated. The functions of these differential metabolites were determined by KEGG pathway enrichment analysis. In Figure 5B, L-valine, L-threonine, L-proline, and L-tyrosine were shown to be enriched in the pathways of aminoacyl-tRNA biosynthesis, phenylalanine, tyrosine and tryptophan biosynthesis, and proline metabolism, while butyric acid and thiamine were found to be mainly enriched in the pathways of butanoate metabolism and thiamine metabolism.

### 3.4. APS Upregulated the Levels of SCFAs in Feces of T2DM Patients

Figure 6A–F show the levels of SCFAs in the three groups detected by GC. The T2DM group was shown to be lower than the HC group in the levels of all SCFAs. The fermentation of APS in the feces of T2DM patients was seen to increase the propionic acid content (*p* < 0.01) and partially recover both acetic and butyric acid levels. However, the three groups showed no significant difference in isobutyric, valeric, and isovaleric acids (Figure 6D–F).

### 3.5. APS Fermentation Enhanced Antioxidant Activity in Feces of T2DM Patients

Oxidative stress is one of the key pathogenic factors that induce T2DM, and the APS antioxidant activity was investigated by analyzing its ability to scavenge DPPH-, hydroxyl radicals (HO•-), and reduction ability in this study. As shown in Figure 6G–I, compared to the T2DM group, the APS-T2DM group was significantly (*p* < 0.05) higher in the scavenging ability of DPPH- and reduction ability, but with no significant difference between them in the scavenging ability of HO•.

### 3.6. Correlations between Gut Microbiota and Differential Metabolites or Oxidative Stress

The results of Spearman correlation analysis between differential metabolites and gut microbiota are shown in Figure 7A,B. *Escherichia-Shigella* was negatively correlated with all-trans-retinoic acid and glutamine, while positively correlated with L-threonine and spermidine. *Lactobacillus* had a positive correlation with thiamin, valeric acid, and DPPH- clearance rate, and a negative correlation with dopamine. *Bifidobacterium* and *Faecalibacterium* exhibited a positive correlation with butyric acid. *Parabacteroides* showed a negative correlation with propionic acid and a positive correlation with spermidine, L-valine, and L-threonine.

## 4. Discussion

Gastrointestinal simulation and in vitro fermentation are commonly used in vitro to study food digestion and influence on the structure of gut microbiota. For example, Hu et al. [29] used gastrointestinal simulation and in vitro fermentation to study the molecular weight changes of extracellular polysaccharides of Corynebacterium parapsilosis during gastrointestinal digestion and their effects on human intestinal microbiota. Yi et al. [30] used gastrointestinal simulation and in vitro fermentation to investigate the digestive characteristics of brown rice gel and its effects on intestinal microbiota, and Xie et al. [31] used gastrointestinal simulation and in vitro fermentation to study the catabolic metabolism of polyphenols of mung bean hull and their effects on intestinal microbiota. Gastrointestinal simulation and in vitro fermentation have the advantages of being fast, cheap, without ethical restrictions, and are widely used. However, these do not fully mimic the in vivo environment and requires in vivo experimental validation compared to human trials [32]. The gastrointestinal simulation results in this paper suggest that APS is barely digested in the gastrointestinal simulation solution, which indicates that it can reach the intestine and be utilized by the gut microbiota without any problem.

Previous studies have confirmed that hyperglycemia and insulin resistance in T2DM patients are strongly associated with gut dysbiosis. High enrichment of conditionally pathogenic bacteria such as *Escherichia coli*, *Helicobacter pylori*, *Salmonella*, and *Vibrio* was reported to induce or promote T2DM development and progression [33,34]. Conversely, commensal or beneficial bacteria can improve glucolipid metabolism and favor health. Firmicutes, represented by Ruminococcaceae, *Clostridium* spp., and Lactobacillaceae, can hydrolyze starch and other sugars to produce butyric acid and other SCFAs, which are often used as probiotics [35]. Studies have also shown that *Bifidobacterium* and *Lactobacillus* can regulate lipid metabolism, lower blood glucose, and reduce T2DM complications [36,37]. In the present study, APS fermentation was shown to increase the abundance of Firmicutes, *Bifidobacterium*, and *Lactobacillus* in T2DM feces and decrease the levels of Proteobacteria, *Escherichia-Shigella*, and *Parabacteroides*. Our results are consistent with a previous report that *Astragalus membranaceus polysaccharides* could improve glucolipid metabolism disorders and alleviate symptoms in T2DM mice by increasing *Allobaculum* and *Lactobacillus* abundance and decreasing *Escherichia coli* abundance [38].

Metabolites produced by intestinal microorganisms have been shown to impact the host’s gut and health [39]. In this study, APS fermentation was observed to significantly reduce the levels of threonine, L-valine, L-threonine, L-proline, and spermidine, and increase the levels of all-trans-retinoic acid, thiamine, glutamine, propanoic acid, and butyric acid in the feces of T2DM patients. This agreed well with the findings in previous studies. For instance, T2DM patients were reported to have higher levels of L-threonine, glutamine, L-valine, L-proline, and spermidine, with these amino acids as potential biomarkers in T2DM patients [40,41]. All-trans-retinoic acid is a biologically active metabolite of retinoic acid that can significantly reduce blood glucose levels in diabetic rats [42]. Thiamine is commonly deficient in diabetic patients and its reduction can impair the endocrine function of the pancreas, thereby exacerbating insulin deficiency and hyperglycemia [43]. Glutamine can prevent or delay T2DM onset by reducing the inflammatory response and promoting insulin sensitivity in skeletal muscle [44]. The differential metabolite-based metabolic pathways are aminoacyl-tRNA biosynthesis, butanoate metabolism, and thiamine metabolism, which are dysregulated in T2DM patients [45,46]. Based on these reports and our results, APS can be concluded to significantly regulate glucolipid metabolism and exert lipid-lowering effects through fecal metabolites and metabolite pathways.

SCFAs are important gut microbiota-generated metabolites and their levels are usually very low in T2DM or obesity patients [47]. SCFAs, especially acetic acid, butyric acid, and propionic acid, have the function of improving glucose regulation and insulin sensitivity in T2DM patients by triggering glucagon-like peptide-1 (GLP-1) and peptide YY (PYY) production, inhibiting β-cell apoptosis, and stimulating insulin secretion [48,49]. In this study, APS was found to upregulate the contents of acetic, propionic, and butyric acids in the stools of T2DM patients, probably due to the reason that APS can promote the proliferation of Firmicutes, *Lactobacillus*, and *Bifidobacterium*. Similarly, oral administration of APS was reported to augment the abundance of *Muribaculum*, *Lactobacillus*, and *Faecalibaculum* and raise the levels of acetic and propionic acids in a diabetic mice model [17].

Chronic oxidative stress can lead to mitochondrial damage, pancreatic β-cell apoptosis, and insufficient insulin secretion [50]. The antioxidant effect of APS has been demonstrated to contribute to the remission and treatment of diabetes [51]. In another study, some antioxidants that are difficult to digest and absorb were found to be fermented by intestinal microorganisms into antioxidant metabolites upon arrival in the intestine, resulting in antioxidant properties in feces [52]. In the present study, the fermentation of APS was found to significantly increase reduction ability and the clearance rate of DPPH- in the feces of T2DM patients, which may be related to the fact that APS can regulate the intestinal microbiota and promote the production of antioxidant metabolites, such as glutamine [53], thiamine [54], and all-trans-retinoic acid [42].

The correlations between intestinal microbiota and differential metabolites or chronic oxidative stress were further confirmed by correlation analysis. Interestingly, *Lactobacillus* was significantly positively associated with thiamine and DPPH- clearance. This is consistent with previous reports that *Lactobacillus* can produce thiamine [55] and have the potential to enhance DPPH- clearance [56]. Therefore, APS can be assumed to increase thiamine levels by promoting *Lactobacillus* content, which in turn improves fecal DPPH- clearance in T2DM patients.

## 5. Conclusions

In this study, the impacts of APS on the intestinal microbiota and its metabolites in T2DM patients were investigated using an in vitro simulated fermentation model. APS addition was shown to improve the imbalance of intestinal microbiota by increasing the abundance of *Bifidobacterium* and *Lactobacillus* and decreasing the abundance of *Escherichia-Shigella*. Additionally, based on metabolomics results, APS could increase all-trans-retinoic acid, thiamine, glutamine, and propanoic acid levels and decrease L-threonine L-valine, L-proline, and spermidine levels, with these metabolites mainly enriched in the pathways of aminoacyl-tRNA biosynthesis, butanoate metabolism, and thiamine metabolism. Moreover, APS could upregulate the levels of acetic, butyric, and propionic acids in the stools of T2DM patients and improve the corresponding antioxidant properties. Furthermore, based on correlation analysis results, *Escherichia-Shigella* was negatively correlated with all-trans-retinoic acid and glutamine while positively correlated with L-threonine and spermidine; *Bifidobacterium* and *Lactobacillus* had a positive correlation with butyric acid and thiamine, respectively. All these results indicate that APS may attenuate type 2 diabetes by modulating gut microbes and metabolites. This study has laid a theoretical foundation for the development of APS as an anti-diabetic drug.

## Figures and Tables

**Figure 1 nutrients-16-01698-f001:**
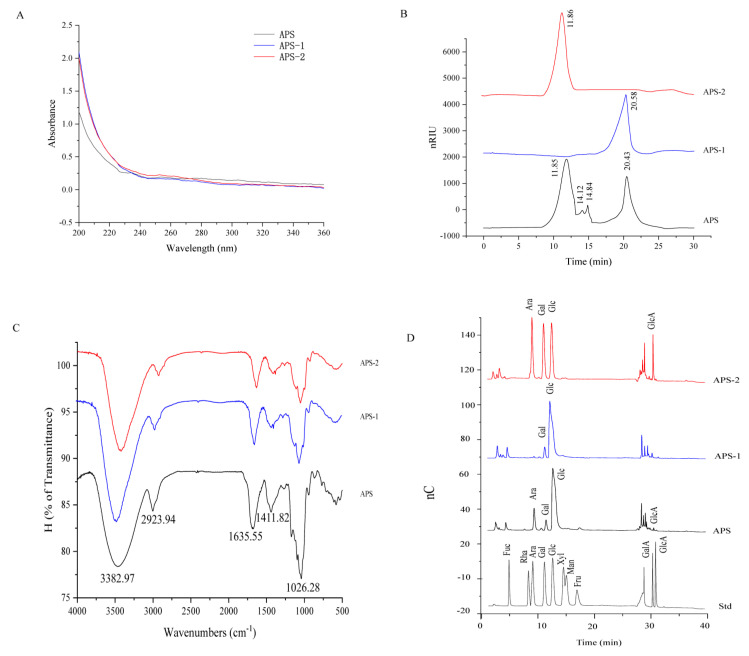
Structure analysis of APS, AP-1, and APS-2. (**A**) UV spectra. (**B**) Molecular weight distribution of APS. (**C**) FT-IR spectra. (**D**) Monosaccharide composition analysis by ion chromatography. Fuc, Fucose; Rha, Rhamnose; Ara, Arabinose; Gal, Galactose; Glc, Glucose; Xyl, Xylose; Man, mannose; Fru, Fructose; GalA, Galacturonic acid; GlcA, Glucuronic acid.

**Figure 2 nutrients-16-01698-f002:**
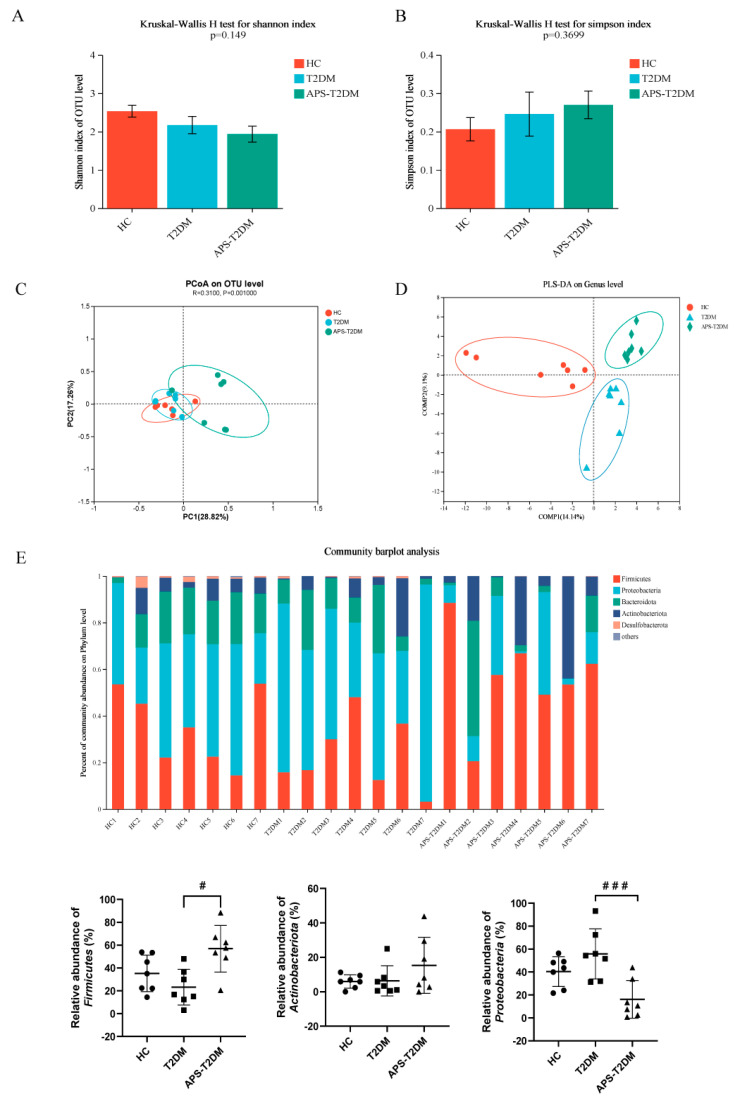
Effects of APS on gut microbiota structure of T2DM patients. (**A**) Shannon index. (**B**) Simpson index. (**C**) PCoA analysis. (**D**) PLS-DA analysis. (**E**) Taxonomic analysis of samples at phylum levels. # *p* < 0.05, ### *p* < 0.01 vs. T2DM group, *n* = 7. Black Circles represent HC, black squares represent T2DM, and black triangles represent APS-T2DM.

**Figure 3 nutrients-16-01698-f003:**
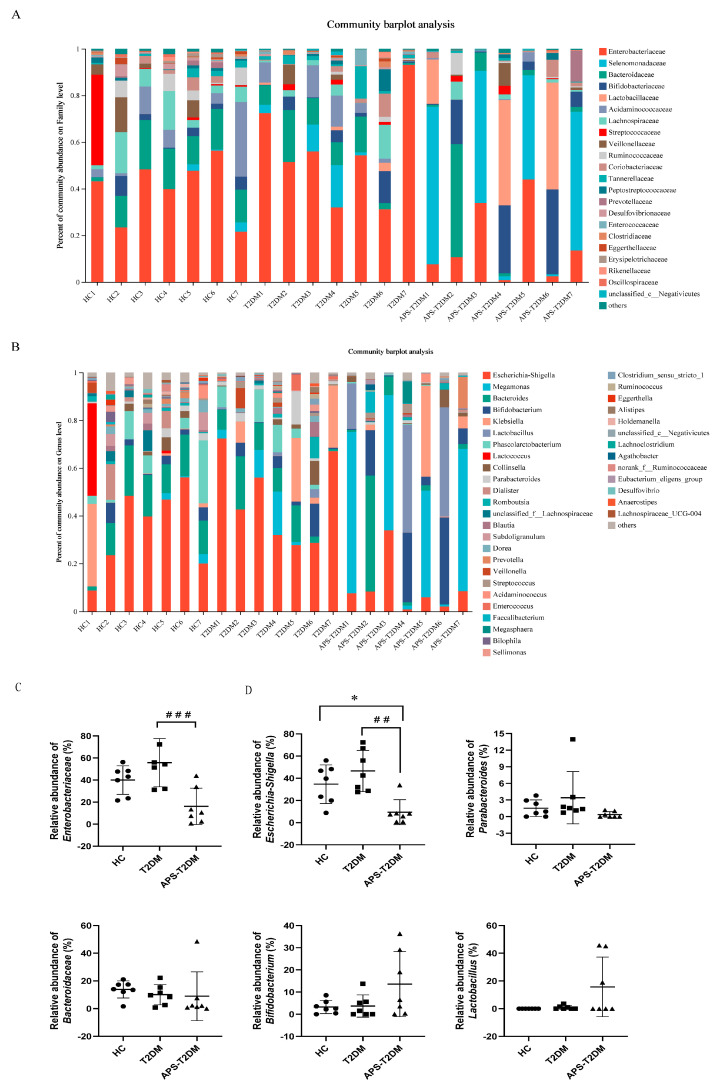
Effects of APS on intestinal microbiota composition in T2DM patients at family and genus levels. (**A**) Gut microbial composition at the family level. (**B**) Gut microbial composition at the genus level. (**C**) Relative abundance analysis of gut microbiota at the family level. (**D**) Relative abundance analysis of gut microbiota at the genus level. ## *p* < 0.01, ### *p* < 0.001 vs. T2DM group, and * *p* < 0.05 vs. HC group, *n* = 7. Black Circles represent HC, black squares represent T2DM, and black triangles represent APS-T2DM.

**Figure 4 nutrients-16-01698-f004:**
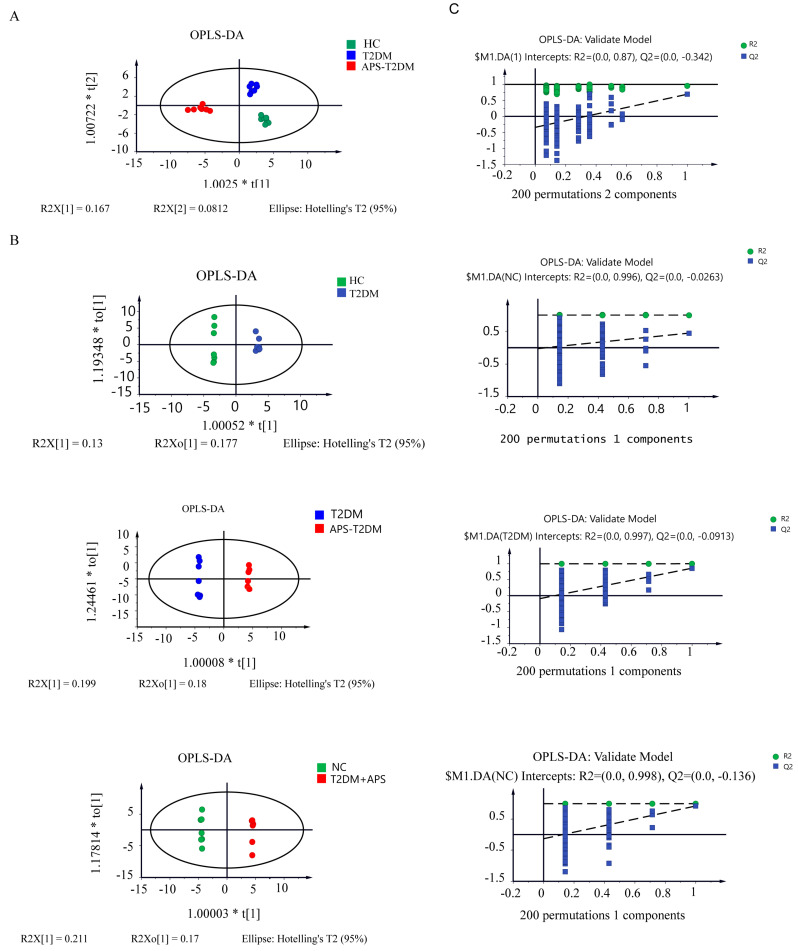
Analyses of metabolites in in vitro fermentation samples by GC-MS and HPLC-MS. (**A**) The OPLS-DA plot of three groups. (**B**) OPLS-DA plots of T2DM vs. HC, APS-T2DM vs. T2DM, and APS-T2DM vs. HC. (**C**) OPLS-DA validation plot (*n* = 200).

**Figure 5 nutrients-16-01698-f005:**
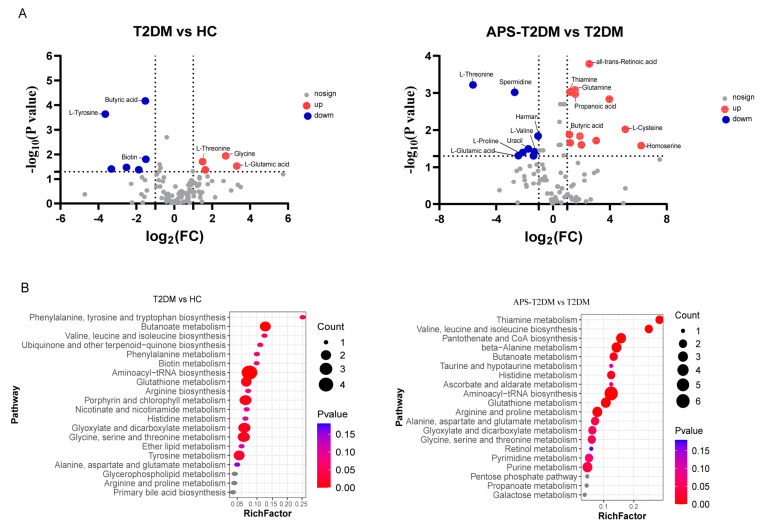
Differential metabolite analysis. (**A**) Volcano plots of metabolites, with blue dots for downregulated metabolites (*p* < 0.05; FC < 0.5 and VIP > 1), red dots for upregulated metabolites (*p* < 0.05; FC > 2 and VIP > 1), and gray dots for metabolites with no significant differences (*p* > 0.05; FC < 2 and VIP < 1). (**B**) KEGG enrichment analysis of different metabolites.

**Figure 6 nutrients-16-01698-f006:**
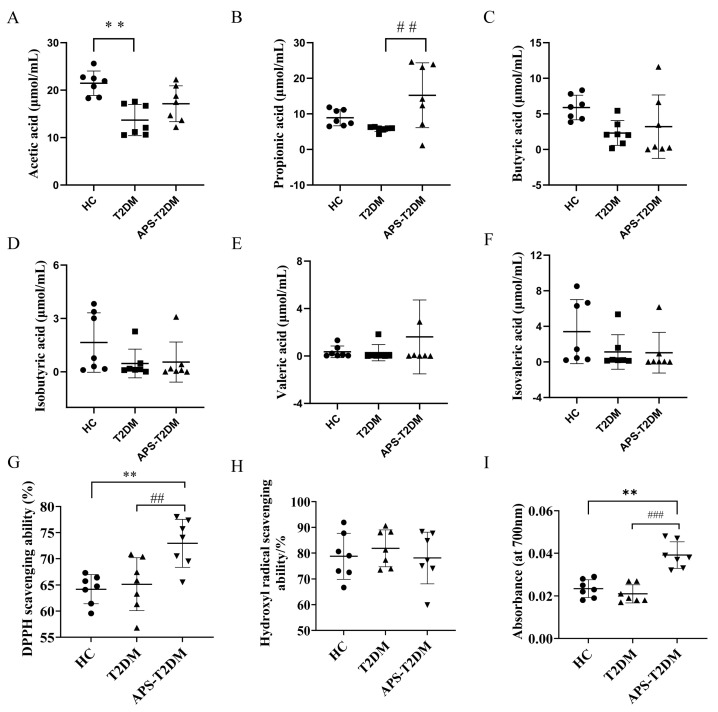
Effects of APS on the levels of SCFAs and antioxidant activities in the feces of T2DM patients after in vitro fermentation. (**A**) Acetic acid. (**B**) Propionic acid. (**C**) Butyric acid. (**D**) Isobutyric acid. (**E**) Valeric acid. (**F**) Isovaleric acid. (**G**) DPPH- scavenging rate. (**H**) Hydroxyl radical scavenging rate. (**I**) Reduction ability. ## *p* < 0.01 vs. ### *p* < 0.001 vs. T2DM group. ** *p* < 0.01 vs. HC group; Black Circles represent HC, black squares represent T2DM, and black triangles represent APS-T2DM. *n* = 7.

**Figure 7 nutrients-16-01698-f007:**
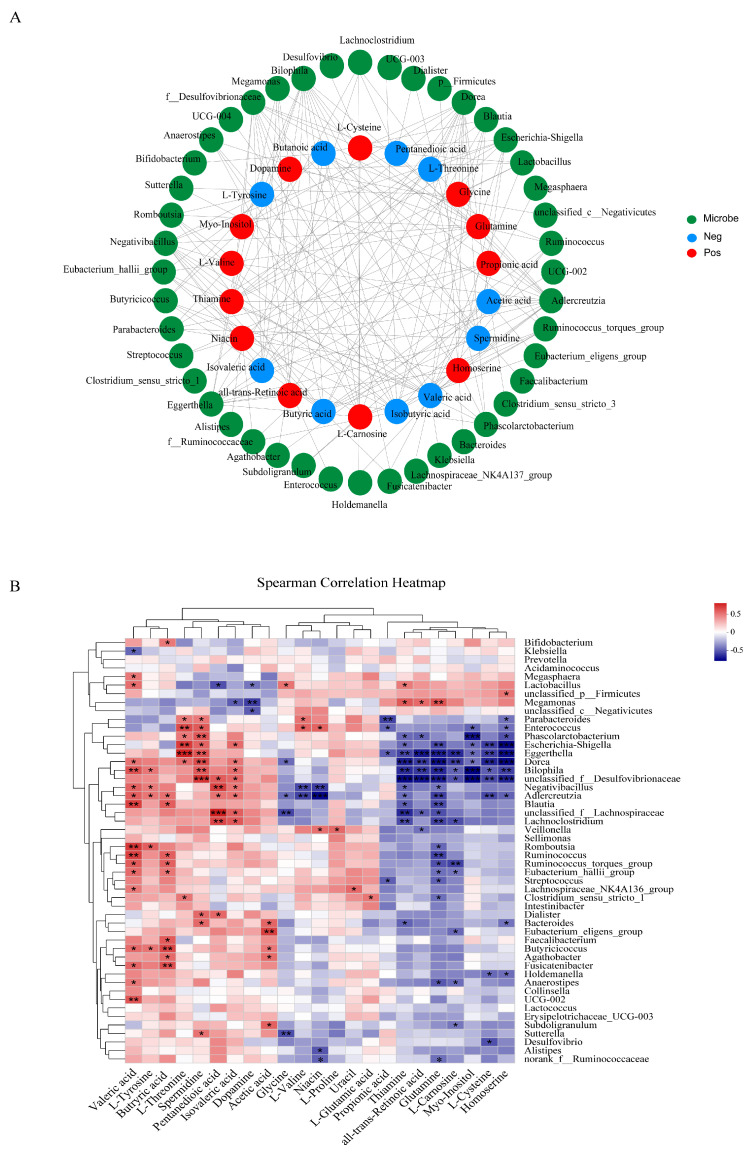
Spearman correlation analysis. (**A**) Network analysis of correlations between metabolites and the top 45 most abundant gut microorganisms at the genus level, with green for microbe, red for positive correlation, and blue for negative correlation. (**B**) Heatmap analysis of correlations between metabolites and microbiota at the genus level, with the color from red to blue in the heatmap representing changes of the R values of Spearman’s correlations from greater to lower. * 0.01 < *p* ≤ 0.05, ** 0.001 < *p* ≤ 0.01, and *** *p* ≤ 0.0001.

## Data Availability

The data for this study are available in the article and the Appendix A.

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
