# Peer review of "Astragalus Polysaccharide Modulates the Gut Microbiota and Metabolites of Patients with Type 2 Diabetes in an In Vitro Fermentation Model"

_nutrients, 2024, doi:10.3390/nu16111698_

Round 1

Reviewer 1 Report

Comments and Suggestions for Authors

This study aims to evaluate the impact of Astragalus polysaccharide (APS), a traditional Chinese medicine, on the gut microbial community shift and metabolite production in T2D patients using an in vitro fermentation system. It is carefully designed to add translational values by using fresh fecal samples from human subjects, incorporating simulations of digestion and anaerobic fermentation. The data are straightforward in the context of the impact of APS fermentation, including the enhanced proliferation of bacteria for SCFA production and modification of thiamine metabolism, leading to enhanced antioxidative capacity. Additionally, the authors clearly demonstrate the correlation analysis of gut metabolite production changes induced by APS with 1) responsive metabolic pathways (KEGG enrichment analysis) and 2) bacterial genus levels (Spearman correlation). Below are some minor comments for the authors:

1.     Although this study targets the therapeutic effects of APS, it would be beneficial to include an APS-HC group to predict the gut microbial community shift in healthy individuals. What are the benefits of APS for healthy humans?

2.     It is recommended to add one of two paragraphs regarding the limitations of the study in simulating GI tract digestion and anaerobic fermentation.

3.     Please describe the differences between the two alpha diversity tests, Shannon and Simpson, and explain why both were employed.

4.     For the beta diversity test in Fig 2C, PCA regression does not seem to separate groups, although the authors claimed clear separation. It displayed better separation by PLS regression. Please briefly explain why PLS exhibits better separation than PCA regression.

5.     Spell out the acronym DPPH.

Author Response

Response to the reviewers’ comments:

Thank you for your letter and for the reviewers’ comments concerning our manuscript entitled " Astragalus polysaccharide modulates the gut microbiota and metabolites of patients with Type 2 diabetes in an in vitro fer-mentation model" (nutrients-2989336). Those comments are all valuable and very helpful for revising and improving our paper, as well as the important guiding significance to our researches. We revised our paper after reading the comments and request by the reviewer and editor. Revised portions are highlighted in yellow in the paper. The following is point by point response to editor and reviewers' comments. If you have any questions, please contact us without hesitate.

1.Comments and Suggestions for Authors

This study aims to evaluate the impact of Astragalus polysaccharide (APS), a traditional Chinese medicine, on the gut microbial community shift and metabolite production in T2D patients using an in vitro fermentation system. It is carefully designed to add translational values by using fresh fecal samples from human subjects, incorporating simulations of digestion and anaerobic fermentation. The data are straightforward in the context of the impact of APS fermentation, including the enhanced proliferation of bacteria for SCFA production and modification of thiamine metabolism, leading to enhanced antioxidative capacity. Additionally, the authors clearly demonstrate the correlation analysis of gut metabolite production changes induced by APS with 1) responsive metabolic pathways (KEGG enrichment analysis) and 2) bacterial genus levels (Spearman correlation). Below are some minor comments for the authors:

  1. Although this study targets the therapeutic effects of APS, it would be beneficial to include an APS-HC group to predict the gut microbial community shift in healthy individuals. What are the benefits of APS for healthy humans?

Respond: Thank you very much for your professional advice. In fact, we initially conducted an experiment of APS fermentation of intestinal flora in healthy group (i.e. APS-HC group), and the results showed that the addition of APS had no significant effect on the α diversity of intestinal flora in healthy group, as shown in Figures 1 A. The results of β diversity indicated that the addition of APS was significantly different from that of HC group, suggesting that APS could affect the intestinal flora structure of HC group (Figures 1 B). Further, the phylum-level results showed that the addition of APS significantly decreased the abundance of Proteobacteria, and the family and genus level results showed that APS significantly decreased enterococcus and had a certain promotion effect on bifidobacteria, but not significantly(Figures 1 and 2.). These results indicated that APS had no significant effect on the abundance of intestinal flora in the healthy group, but had a positive regulatory effect on the intestinal flora in the healthy group, such as inhibiting pathogenic bacteria and increasing the abundance of beneficial bacteria. Finally, considering that the research focus of this paper is mainly to investigate the impact of APS on the intestinal flora of diabetic patients, the experimental results of the impact of APS on the healthy group have not been presented.

Figure 1. Effects of APS on gut microbiota structure of T2DM patients. (A) Shannon index. (B) Simpson index. (C) PCoA analysis. (D) PLS-DA analysis. (E) Taxonomic analysis of samples at phylum levels. #p<0.05, ### p<0.01 vs. T2DM group, n=7.

Figure 2. Effects of APS on intestinal microbiota composition in T2DM patients at family and ge-nus levels. (A) Gut microbial composition at the family level. (B) Gut microbial composition at the genus level. (C) Relative abundance analysis of gut microbiota at the family level. (D) Relative abundance analysis of gut microbiota at the genus level. # #p < 0.01, ###p < 0.001 vs. T2DM group, and *p < 0.05 vs. HC group, n = 7

  1. It is recommended to add one of two paragraphs regarding the limitations of the study in simulating GI tract digestion and anaerobic fermentation.

RespondThank you for your professional question. Gastrointestinal simulation and in vitro fermentation are commonly used methods to study food digestion and influence on the structure of the flora. For example, Hu et al[1] used gastrointestinal simulation and in vitro fermentation to study the molecular weight changes of extracellular polysaccharides of Sporobacterium parvum during gastrointestinal digestion and their effects on human intestinal microbiota.Yi [2] et al. used gastrointestinal simulation and in vitro fermentation to investigate the digestive properties of brown rice gel and the effects on intestinal microbiota. Xie et al.[3] used gastrointestinal simulation and in vitro fermentation to study the catabolism of polyphenols of mungbean skin and their effects on intestinal flora. effects on intestinal microbiota. Although gastrointestinal simulation and in vitro fermentation have the advantages of being rapid, inexpensive, without ethical restrictions and widely used. However, in vitro digestion and fermentation do not fully reflect the real situation of food digestion and fermentation in the human body compared to human trials, and the results need to be verified by in vivo experiments. We have added the relevant discussion to the first paragraph of the discussion。

[1] Hu B , Liu C , Jiang W ,et al.Chronic in vitro fermentation and in vivo metabolism: Extracellular polysaccharides from Sporidiobolus pararoseus regulate the intestinal microbiome of humans and mice[J].International Journal of Biological Macromolecules, 2021, 192:398-406.DOI:10.1016/j.ijbiomac.2021.09.127.

[2] Yi C , Xu L , Luo C ,et al.In vitro digestion, fecal fermentation, and gut bacteria regulation of brown rice gel prepared from rice slurry backfilled with rice bran[J].Food hydrocolloids, 2022.

[3] Xie, J.; Sun, N.; Huang, H.; Xie, J.; Chen, Y.; Hu, X.; Hu, X.; Dong, R.; Yu, Q. Catabolism of polyphenols released from mung bean coat and its effects on gut microbiota during in vitro simulated digestion and colonic fermentation. Food Chem 2022, 396, 133719, doi:10.1016/j.foodchem.2022.133719.

  1. Please describe the differences between the two alpha diversity tests, Shannon and Simpson, and explain why both were employed.

Respond:Shannon's index is used to describe the disorder and uncertainty in the occurrence of individuals of a species, the higher the uncertainty, the higher the diversity. Simpson's Diversity Index reflects the size of species richness by analyzing the number of individuals in the same population. the value of this index ranges from 0 to 1, with larger values indicating lower species identification and higher concentration of species numbers. both Shannon and Simpson indices can be used to measure species diversity. However, these two indices are calculated in different ways and with different emphasis, so we generally look at them together to avoid discrepancies between the results of the two indices. Similarly, Liu et al. [1] and Zhang et al. [2] used Shannon and Simpson indices to characterize the diversity of gut microbiota.

[1] Liu, W.; Li, X.; Zhao, Z.; Pi, X.; Meng, Y.; Fei, D.; Liu, D.; Wang, X. Effect of chitooligosaccharides on human gut microbiota and antiglycation. Carbohydr Polym 2020, 242, 116413, doi:10.1016/j.carbpol.2020.116413.

[2] Zhang, X.; Wang, H.; Xie, C.; Hu, Z.; Zhang, Y.; Peng, S.; He, Y.; Kang, J.; Gao, H.; Yuan, H., et al. Shenqi compound ameliorates type-2 diabetes mellitus by modulating the gut microbiota and metabolites. J Chromatogr B Analyt Technol Biomed Life Sci 2022, 1194, 123189, doi:10.1016/j.jchromb.2022.123189.

  1. For the beta diversity test in Fig 2C, PCA regression does not seem to separate groups, although the authors claimed clear separation. It displayed better separation by PLS regression. Please briefly explain why PLS exhibits better separation than PCA regression.

Respond Thank you for your comments, there is no PCA in this paper, only PCoA. Similar to PCA, PCoA finds the most dominant coordinates in the distance matrix by sorting through a series of eigenvalues and eigenvectors and then selecting the eigenvalues that are mainly ranked in the top positions. PCoA is a non-constrained method of analyzing the dimensionality of the data downscaling, i.e., each sample has the same contribution to the model. For samples with insignificant differences between groups, PCA does not show the inter-group variability well. PLS-DA is a supervised discriminant method for samples with insignificant between-group differences. PLS-DA uses partial least squares regression to model the relationship between variables and sample categories to better analyze and visualize between-group differences in samples. We hope you find our answer satisfactory.

  1. Spell out the acronym DPPH.

Respond Thanks to your suggestion, we have added 2,2-diphenyl-1-picrylhydrazyl (DPPH) to the corresponding position in the manuscript.

Reviewer 2 Report

Comments and Suggestions for Authors

Dear authors,

the manuscript "Astragalus polysaccharide modulates the gut microbiota and metabolites of patients with Type 2 diabetes in an in vitro fermentation model" I was reviewing interested me very much. Interesting subject matter, presentation of the topic legible.  I have a suggestion for the improvement of the current manuscript  before publishing:

1. MTT or other assays on immortalized human intestinal epithelial cells should be used to demonstrate the absence of APS' cytotoxicity.  This an important information in order to validate your results.

Best regards

Author Response

Response to the reviewers’ comments:

Thank you for your letter and for the reviewers’ comments concerning our manuscript entitled " Astragalus polysaccharide modulates the gut microbiota and metabolites of patients with Type 2 diabetes in an in vitro fer-mentation model" (nutrients-2989336). Those comments are all valuable and very helpful for revising and improving our paper, as well as the important guiding significance to our researches. We revised our paper after reading the comments and request by the reviewer and editor. Revised portions are highlighted in yellow in the paper. The following is point by point response to editor and reviewers' comments. If you have any questions, please contact us without hesitate.

  1. Comments and Suggestions for Authors

Dear authors,

the manuscript "Astragalus polysaccharide modulates the gut microbiota and metabolites of patients with Type 2 diabetes in an in vitro fermentation model" I was reviewing interested me very much. Interesting subject matter, presentation of the topic legible.  I have a suggestion for the improvement of the current manuscript before publishing:

  1. MTT or other assays on immortalized human intestinal epithelial cells should be used to demonstrate the absence of APS' cytotoxicity.  This an important information in order to validate your results.

Respond Thank you for your valuable comments, according to your request, we examined the effect of APS on the activity of Caco2 cells, the results are shown in the figure below, the concentration of APS at 50-400µg/mL has no significant effect on the cell activity. Consistent with my results Ying et al. [1] and Wang et al. [2] also found that APS was not toxic to Caco2 cells. Meanwhile, we added this part of the results to the Results and Supplementary file.

 [1] Ying, Y.; Song, L.Y.; Pang, W.L.; Zhang, S.Q.; Yu, J.Z.; Liang, P.T.; Li, T.G.; Sun, Y.; Wang, Y.Y.; Yan, J.Y., et al. Astragalus polysaccharide protects experimental colitis through an aryl hydrocarbon receptor-dependent autophagy mechanism. Br J Pharmacol 2024, 181, 681-697, doi:10.1111/bph.16229.

[2] Wang, X.; Li, Y.; Yang, X.; Yao, J. Astragalus polysaccharide reduces inflammatory response by decreasing permeability of LPS-infected Caco2 cells. Int J Biol Macromol 2013, 61, 347-352, doi:10.1016/j.ijbiomac.2013.07.013

Supplementary Figure 2. The effects of APS on the viability of Caco-2 cells for 24 h in different concentrations (50, 100, 200, 400 μg/mL). Cell viability was determined using the MTT assay.

Reviewer 3 Report

Comments and Suggestions for Authors

Dear Authors,

I find the manuscript submitted to Nutrients, titled "Astragalus polysaccharide modulates the gut microbiota and metabolites of patients with Type 2 diabetes in an in vitro fermentation model," scientifically intriguing and potentially impactful for further research concerning the management of lifestyle diseases such as Type 2 diabetes. However, I have several points for consideration:

  1. The introduction seems lacking in depth with only 12 citations, which may not sufficiently contextualize existing literature and demonstrate the significance of the current study.
  2. In the Materials and Methods section (paragraph 2.1), it is unclear when the plant material was collected and when the experiments were conducted.
  3. Lines 75-76: Why was the extract concentrated from an aqueous solution using rotary evaporation rather than lyophilization?
  4. Line 140: What was the concentration of the ribitol solution used as a standard?
  5. In Section 3.1, a table detailing the composition of APS fractions would aid in analyzing the extract compositions. Additionally, were quantitative analyses of individual components conducted?
  6. The analysis of the study is somewhat challenging due to the volume of data and the need to navigate between the article and the supplement. Listing the abbreviations used may facilitate comprehension.

Author Response

Response to the reviewers’ comments:

Thank you for your letter and for the reviewers’ comments concerning our manuscript entitled " Astragalus polysaccharide modulates the gut microbiota and metabolites of patients with Type 2 diabetes in an in vitro fer-mentation model" (nutrients-2989336). Those comments are all valuable and very helpful for revising and improving our paper, as well as the important guiding significance to our researches. We revised our paper after reading the comments and request by the reviewer and editor. Revised portions are highlighted in yellow in the paper. The following is point by point response to editor and reviewers' comments. If you have any questions, please contact us without hesitate.

  1. Comments and Suggestions for Authors

Dear Authors,

I find the manuscript submitted to Nutrients, titled "Astragalus polysaccharide modulates the gut microbiota and metabolites of patients with Type 2 diabetes in an in vitro fermentation model," scientifically intriguing and potentially impactful for further research concerning the management of lifestyle diseases such as Type 2 diabetes. However, I have several points for consideration:

  1. The introduction seems lacking in depth with only 12 citations, which may not sufficiently contextualize existing literature and demonstrate the significance of the current study.

RespondThank you for your professional opinion, and based on your suggestions, we have added relevant literature and discourse to the background introduction. Your comments have improved our article.

  1. In the Materials and Methods section (paragraph 2.1), it is unclear when the plant material was collected and when the experiments were conducted.

Respond Thank you for your professionalism, Astragalus is purchased from origin Min County, Dingxi City, Gansu Province. Like us, the astragalus studied by Yang et al. [1] and Wang et al. [2] was also purchased directly. Generally speaking, after 2-3 years of growth, astragalus is harvested in late October, then dried, tied into small bundles, and can be used.

[1] Yang B , Xiong Z , Lin M ,et al.Astragalus polysaccharides alleviate type 1 diabetes via modulating gut microbiota in mice.[J].International journal of biological macromolecules, 2023:, 123767. DOI:10.1016/j.ijbiomac.2023.123767.

[2] Astragalus polysaccharides decreased the expression of PTP1B through relieving ER stress induced activation of ATF6 in a rat model of type 2 diabetes[J].Molecular & Cellular Endocrinology, 2009, 307(1-2):89-98.DOI:10.1016/j.mce.2009.03.001.

  1. Lines 75-76: Why was the extract concentrated from an aqueous solution using rotary evaporation rather than lyophilization?

RespondThank you for your comments. When we extract polysaccharides, we need to soak them with a lot of hot water. Compared with freeze-drying method, rotary evaporation has the advantages of fast, high efficiency and simple operation. In order to improve the efficiency we chose the rotary evaporation method. Similarly, Zhong et al. [1] and Zhou et al. [2] also concentrated extracts by rotary evaporation.

[1] Zhong K , Fan S , Yao S ,et al.A Atractylodes lancea polysaccharide inhibits metastasis of human osteosarcoma U‐2 OS cells by blocking sialyl Lewis X (sLex)/E‐selectin binding[J].Journal of Cellular and Molecular Medicine, 2020, 24(21).DOI:10.1111/jcmm.15870.

[2] Zhou, P.; Xiao, W.; Wang, X.; Wu, Y.; Zhao, R.; Wang, Y. A Comparison Study on Polysaccharides Extracted from Atractylodes chinensis (DC.) Koidz. Using Different Methods: Structural Characterization and Anti-SGC-7901 Effect of Combination with Apatinib. Molecules 2022, 27, doi:10.3390/molecules27154727.

  1. Line 140: What was the concentration of the ribitol solution used as a standard?

RespondThanks for your comments, we have added the concentration of ribosol (1 mg/mL) to the corresponding material method.

  • In Section1, a table detailing the composition of APS fractions would aid in analyzing the extract compositions. Additionally, were quantitative analyses of individual components conducted

RespondThank you for your comments. In fact, we mentioned in the results of section 3.1 that the ratios of the three components APS-0m, APS-0.1m and APS-0.2m to the total sugars of APS were 82.72 ± 4.10%, 2.65 ± 0.01% and 13.14 ± 0.02%, respectively. For ease of observation, we have added the proportion values to supplementary Figure 1A. In addition, we analyzed the chemical composition, monosaccharide composition, and relative molecular mass of purified APS polysaccharides APS-1 and APS-2 (as shown in supplementary Table 2). Finally, although presenting the percentage of each component of APS in a table would be a clearer and more explicit result. However, considering the large number of images in this paper, this part of the data was added in the supplementary images. We hope you will find our answer satisfactory.

Supplementary Figure 1 Purification of APS. (A) Purification procedure of APS. (B) Stepwise elution curve of APS on DEAE Sepharose FF chromatography column. Elution curve of APS-1(C) and APS-2 (D) on Sephadex G-100 chromatography column.

Supplementary Table 2 Basic physicohemical properties and structure analysis of APS

6.The analysis of the study is somewhat challenging due to the volume of data and the need to navigate between the article and the supplement. Listing the abbreviations used may facilitate comprehension.

RespondThank you for your valuable comments, we have added abbreviations to the manuscript.

Round 2

Reviewer 2 Report

Comments and Suggestions for Authors

Dear Authors,

now your manuscript "Astragalus polysaccharide modulates the gut microbiota and metabolites of patients with Type 2 diabetes in an in vitro fermentation model" can be accept in this form.

Best regards

Author Response

Thank you for your recognition of this study.
